# In situ electron paramagnetic resonance spectroscopy using single nanodiamond sensors

Zhuoyang Qin[1,2,6], Zhecheng Wang[1,2,6], Fei Kong [1,2,3] ✉, Jia Su[1,2], Zhehua Huang[1,2], Pengju Zhao[1,2], Sanyou Chen[1,2,4], Qi Zhang [1,2,4], Fazhan Shi [1,2,3,4] ✉ & Jiangfeng Du [1,2,3,5] ✉

An ultimate goal of electron paramagnetic resonance (EPR) spectroscopy is to analyze molecular dynamics in place where it occurs, such as in a living cell. The nanodiamond (ND) hosting nitrogen-vacancy (NV) centers will be a promising EPR sensor to achieve this goal. However, ND-based EPR spectroscopy remains elusive, due to the challenge of controlling NV centers without well-defined orientations inside a flexible ND. Here, we show a generalized zero-field EPR technique with spectra robust to the sensor's orientation. The key is applying an amplitude modulation on the control field, which generates a series of equidistant Floquet states with energy splitting being the orientation-independent modulation frequency. We acquire the zero-field EPR spectrum of vanadyl ions in aqueous glycerol solution with embedded single NDs, paving the way towards in vivo EPR.

Electron paramagnetic resonance (EPR) spectroscopy is a well-established technique for analyzing molecules containing unpaired electrons. Among its widespread applications in diverse scientific fields[1–3], a featured one is the study of dynamic processes, such as monitoring redox reactions[4] and unraveling molecular motions[5]. Performing those studies in living cells is an active research topic[6–8], where an ultimate goal is promoting the EPR detection to single-cell level. Towards this goal, an essential but unmet precondition is developing suitable EPR sensors with both high spin sensitivity and good biocompatibility. The conventional EPR sensor is a macroscopic resonant microwave cavity with limited spin sensitivity. In the past decades, numerous microscopic EPR sensors have been developed to improve the spin sensitivity, including magnetic resonance force microscopy[9], scanning tunneling microscopy[10], and superconducting microresonator[11,12], but they require cryogenic temperatures and high vacuum environments. Alternatively, nitrogen-vacancy (NV) centers in diamond can also serve as EPR sensors with single-spin sensitivity[13,14],

even at ambient conditions[15–17]. Furthermore, the diamond hosting NV centers can shrink to nanometer size, making itself more flexible to be in situ sensor, such as magnetometry inside polymers[18], relaxometry in lipid bilayer[19], and thermometry in living cells[20]. However, using this flexible nanodiamond (ND) as an EPR sensor remains challenging.

The flexibility of NDs, on the other hand, also brings the uncertainty of their orientations, which prevents the hosted NV centers from measuring EPR spectra. It is because the NV center has an anisotropic response to magnetic fields with a principal axis along the N-V axis[21]. In the presence of external static or oscillating magnetic field, random tumbling of the ND will lead to variations in the transition frequency or strength of the hosted NV center, and thus prevents the current EPR detection schemes. For instance, both the double electron-electron resonance (DEER)[15–17,22,23] and the cross-relaxation schemes[24,25] require precise quantum controls of the NV spin states. To overcome this challenge, an active approach is either manipulating the ND orientation, such as, by using optical tweezers[26], tracking the ND orientation[27]

[1]CAS Key Laboratory of Microscale Magnetic Resonance and School of Physical Sciences, University of Science and Technology of China, Hefei 230026, China. [2]CAS Center for Excellence in Quantum Information and Quantum Physics, University of Science and Technology of China, Hefei 230026, China. [3]Hefei National Laboratory, University of Science and Technology of China, Hefei 230088, China. [4]School of Biomedical Engineering and Suzhou Institute for Advanced Research, University of Science and Technology of China, Suzhou 215123, China. [5]School of Physics, Zhejiang University, Hangzhou 310027, China. [6]These authors contributed equally: Zhuoyang Qin, Zhecheng Wang. ✉e-mail: kongfei@ustc.edu.cn; fzshi@ustc.edu.cn; djf@ustc.edu.cn

and then adjusting the control field, or optimizing the control pulses[28]. Besides, another passive but technically simpler way is to develop detection schemes that are naturally insensitive to the orientation. An illuminating example is the zero-field EPR[29,30], where the resonance frequency does not depend on the spin target's orientation, although still depend on the NV sensor's orientation.

Here, we generalize the robustness of zero-field EPR technique to not only the target's but also the sensor's orientations. By applying an extra amplitude modulation on the control field, the modulation frequency, rather than the field strength, will determine the resonance condition. We experimentally demonstrate it by performing the EPR measurements on P1 centers with different driving field strength, showing that the peak position is indeed field-strength independent. To further show the robustness of our method, we immerse the ND in an aqueous glycerol solution of vanadyl sulfate, and then use the hosted NV centers to detect the vanadyl ions. Although both the ND and the ions are tumbling, we can still acquire a clear EPR spectrum. Our results show the possibility of using flexible NDs as EPR sensors to enable in situ and even in vivo EPR measurements.

## Results

### Scheme of zero-field ND-EPR

Considering that a ND is placed inside a sample containing paramagnetic molecules, where all of them are tumbling randomly (Fig. 1a), the task is using the NV center inside the ND to measure the EPR spectrum of the molecules. In the absence of external static magnetic field, the energy levels of both the sensor and the target are irrelevant to their orientations[29]. The Hamiltonian of this sensor-target system can be written as

$$H_0 = DS_z^2 + d_{ij}S_iT_j + \omega T_z, \tag{1}$$

where **S** and **T** are the spin operator of the NV sensor and the spin target, respectively, $d_{ij}$ ($i,j=x,y,z$) is the dipole-dipole coupling

between them, $D = 2.87$ GHz is the zero-field splitting of NV sensor, and $\omega$ is the energy splitting of the target spin induced by the intrinsic interaction.

There usually exists a large energy mismatch between the sensor and the target ($|D - \omega| \gg d$), and thus the dipole-dipole coupling between them has undetectable influence on the sensor. A driving field can eliminate this energy mismatch by bringing the NV center from lab frame to dress frame (Fig. 1b). A direct way is applying a resonant microwave (MW) field $B_1 \cos Dt$ (Fig. 1c)[30], and then we have

$$H_1 = \frac{\Omega}{2}S_x + d_{zj}S_zT_j + \omega T_z \tag{2}$$

in the interaction picture, where $\Omega = \gamma_{NV}B_1 \sin \theta$ is the Rabi frequency, $\gamma_{NV} = -28.03$ GHz/T is the gyromagnetic ratio of the NV electron spin, and $\theta$ is the angle between the MW magnetic field $\mathbf{B_1}$ and the N-V axis. The energy gap of the sensor becomes $\Omega/2$. By sweeping $\Omega$, a resonant cross-relaxation process will happen when $\Omega/2 = \omega$, resulting in a reduction of the photoluminescence (PL) rate of the NV center. Experimentally, the sweep is performed on $B_1$ rather than $\Omega$, so the actual resonance condition is

$$\gamma_{NV}B_1 \sin \theta = \omega. \tag{3}$$

Figure 1d gives a simulation of the dependence of spectra on $\theta$. One can see the peak position varies dramatically when $\theta$ deviates from $\pi/2$. For a randomly tumbling ND, the spectrum will show line broadening and become asymmetric. Note that the two $\pi/2$ pulses in Fig. 1c will also deteriorate, resulting in weaker signal.

To address this issue, we perform an periodical amplitude modulation on the continuous driving MW of the form $B_1 \cos ft \cos Dt$ (Fig. 1e), and then the Hamiltonian Eq. (2) turns into (Supplementary

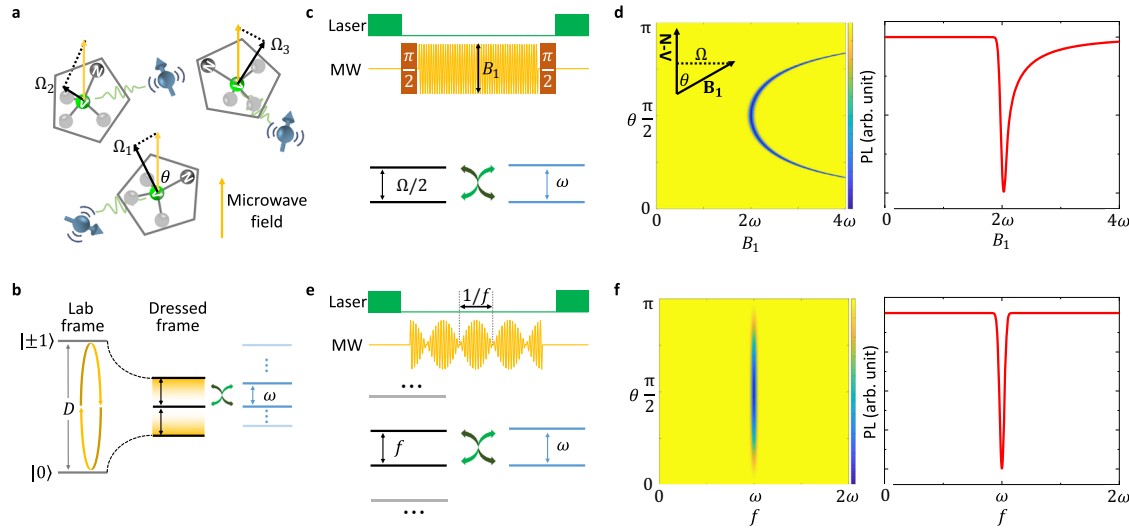

**Fig. 1 | Methods for EPR measurements based on tumbling NDs. a** Simplified model of the ND sensor and the target spin. A microwave field marked by the yellow arrow is applied to control the spin state of the NV center, where only the component perpendicular marked by the black arrows ($\Omega_{1,2,3}$) to the N-V axis matters. **b** Generalized Hartmann-Hahn scheme. The driving field can transfer the NV center from lab frame to dressed frame in order to eliminate the huge energy mismatch between the NV center and the target spins, where $D$ is the zero-field splitting of the NV center and $\omega$ is the energy splitting of the target spin. **c** Pulse sequence and

corresponding energy match condition for direct drive. The black arrows mark the scanning variables, microwave (MW) amplitude $B_1$. **d** Simulated EPR spectra for direct drive. Left side is a simulation of the spectral dependence on $\theta$, while right side is the expected spectra after average over random $\theta$. We omit the gyromagnetic ratio $\gamma_{NV}$ for simplicity. PL, photoluminescence. **e** Pulse sequence and corresponding energy match condition for amplitude-modulated drive. The black arrows mark the scanning variables, amplitude-modulation frequency $f$. **f** Simulated EPR spectra for amplitude-modulated drive.

Note 1)

$$H_{\mathrm{II}} = \sum_{m=\mathrm{odd}} 2J_m\left(\frac{\kappa}{2}\right) d_{zj} \sin mft\, S_y T_j + \omega T_z, \qquad (4)$$

where $J_m$ is the $m$-th order Bessel functions of the first kind, and $\kappa = \Omega/f$ is the relative driving index. This periodical modulation creates a series of Floquet side bands with splitting determined by the modulation frequency $f$. In the situation of $\Omega \ll f$, only the first-order term matters. The dipolar coupling will induce an additional longitudinal relaxation on the NV sensor with rate of (Supplementary Note 1)

$$\Gamma'_1 = \frac{3\kappa^2}{64}\frac{\left(d_{zx}^2 + d_{zy}^2\right)\Gamma_2}{\Gamma_2^2 + (f-\omega)^2}, \qquad (5)$$

where $\Gamma_2 = \Gamma_{2,\mathrm{NV}} + \Gamma_{2,\mathrm{tar}}$ is the total decoherence rate. After a evolution time of $t$, the signal contrast will be

$$S = \frac{2}{3}e^{-\Gamma_1 t}\left(1 - e^{-\Gamma'_1 t}\right), \qquad (6)$$

where $\Gamma_1 = 1/T_1$ is the intrinsic relaxation rate of the NV sensor. So the resonance condition becomes

$$f = \omega. \qquad (7)$$

The energy mismatch can also be removed at the price of a reduced coupling by a factor of $\kappa/4$. Now the resonance condition does NOT depend on $\theta$, but the signal strength does (Fig. 1f). The tumbling induced line broadening is completely removed.

## EPR measurements with fixed NDs
To better see the dependence of EPR spectra on the Rabi frequency $\Omega$, we first perform the experimental demonstrations on fixed NDs, where $\Omega$ is well-defined and adjustable. We place the NDs on a coverslip by spin coating, and observe them on a home-built confocal microscope (Fig. 2a). The 532 nm green laser and the red fluorescence are used to polarized and read out the spin state of NV centers, respectively. Figure 2b shows the Rabi oscillation of the NV centers in one ND. Since each ND contains an average of 12–14 NV centers (see Methods), the Fourier transform of the Rabi oscillation shows different peaks, corresponding to differently oriented NV centers. Here $\Omega$ is defined as the dominated Rabi frequency.

By applying the pulse sequence given in Fig. 1e, we can directly get the zero-field EPR spectrum of P1 centers (Fig. 2c), which are another kind of defects in diamond. Since the signal strength depends on the relative driving index $\kappa$ (Eq. (5) and Eq. (6)), which is proportional to $B_1/f$, we sweep the driving amplitude $B_1$ accordingly during the sweeping of $f$, and keep $\kappa$ as a constant. According to previous measurements on bulk diamonds[30], the zero-field EPR spectrum of [14]N P1 centers should have three peaks at 18 MHz, 130 MHz, and 148 MHz. The first peak is difficult to observe for NDs, because it merges with the broad background peak around $f = 0$ MHz (Fig. 2c). Here we focus on the last two. As shown in Fig. 2c, two clear peaks appear at the expected positions, even though the ND contains several differently oriented NV centers with different effective driving amplitudes. Besides, we repeat the measurement with different $\kappa$ to simulate the rotation of NDs. The peak position indeed shows independence on $\kappa$ (Fig. 2d), while the peak contrasts show positive dependence on $\kappa$ (Fig. 2e), consisting with the theoretical prediction. Therefore, it is promising that the ND-EPR spectrum will be robust to the tumbling of NDs.

## EPR measurements with tumbling NDs
We then perform the measurement on tumbling NDs to directly show the robustness of our scheme. For a free ND in aqueous solutions, it has both rotational and transnational diffusion. To keep the ND staying at the focus of laser, extra techniques such as wide-field excitation and charge-coupled device (CCD) detection[31] or real-time tracking[32] are required, which are beyond the scope of this work. In order to simplify the measurement, we use a soft 'string', which is a polyethylene glycol (PEG) molecule, to tether the ND on the surface of a coverslip (see Methods). Its length is roughly 120 nm, much larger than the ND diameter (~40 nm) and smaller than the laser spot (~800 nm). So the rotational motion is nearly unperturbed, while the transnational motion is restricted. We use a short mPEG molecule to control the density of NDs, so that single NDs can be observed (Supplementary Note 2). We put a glycerol aqueous solution (glycerol:water = 9:1) of vanadyl sulfate with a concentration of 25 mM on the ND-tethered coverslip (see Methods), and then use the embedded ND to sensing the vanadyl ions (Fig. 3a). Here we use the glycerol-water mixture rather than the pure water to reduce the rotational diffusion rate of vanadyl ions, which we will discuss below. As shown in Fig. 3b, the Rabi oscillation decays quickly, corresponding to a wide distribution of Rabi frequency. The oscillation also changes in time slowly, confirming the tumbling of NDs in the aqueous solution. The irregular oscillation shows the difficulty of precisely controlling the spin in a tumbling ND. Nevertheless, we can still clearly acquire the zero-field EPR spectrum of vanadyl ions.

The ion exists as $[\mathrm{VO(H_2O)_5}]^{2+}$ in aqueous solution at moderately low pH[33]. It consists of a electron spin $S = 1/2$ and a nuclear spin $I = 7/2$ with hyperfine interaction between them. At zero magnetic field, the spin Hamiltonian is

$$H_{\mathrm{VO}} = A_\perp\left(S_x I_x + S_y I_y\right) + A_\parallel S_z I_z, \qquad (8)$$

where $A_\perp = 208.5$ MHz, $A_\parallel = 547$ MHz[33], and we neglect the small nuclear quadrupole coupling term. The eigenstates can be written as $|T, m_T\rangle$ ($T = 4, 3, m_T = \pm T, \pm(T-1), ..., 0$), where $\mathbf{T} = \mathbf{S} + \mathbf{I}$ is the total angular momentum. The eigenenergies are $\left(-A_\parallel \pm \sqrt{m_T^2 A_\parallel^2 + (16 - m_T^2)A_\perp^2}\right)/4$, where plus and minus correspond to $T = 4$ and 3, respectively. Due to the axial symmetry ($A_x = A_y = A_\perp$), all the $m_T \neq 0$ states are doubly degenerate, and thus 16 states occupy 9 energy levels (Fig. 3c). In general, all the transitions that meet the selection rule $\Delta m_T = 0, \pm 1$ are observable. For each transition, the vanadyl ion can be simplified to a two-level system $\omega T_z$ as in Eq. (1) with a modified dipolar coupling to the NV sensor (Supplementary Note 3).

Figure 3d gives a simulated EPR spectrum of vanadyl ions, where up to 12 peaks are observable. However, Peaks 5-8 will overlap with the strong signal of P1 centers, and thus be hardly to resolve. There also exists a strong background peak at $D/2 = 1435$ MHz. Because the amplitude-modulated driving field can be divided into two microwaves with frequencies of $D - f$ and $D + f$, each of which alone can also be used for the EPR measurement. This off-resonant driving field is technically simpler, but will induce a second-order shift of the resonance frequency depending on the relative driving index $\kappa$ (Supplementary Note 4). In the case of poor spectral resolution, the off-resonant drive is similar to the amplitude-modulate drive. When $f \approx D/2$, the driving field itself will be an very strong artificial signal (Supplementary Note 4). Besides, observations of higher-frequency peaks require stronger driving power. Therefore, our measurement focus on the middle range, which is enough to extract the hyperfine constants. As shown in Fig. 3d, three clear peaks appear at 780 MHz, 950 MHz, and 1150 MHz, possibly corresponding to peak 1, 2, and 10, respectively. Peak 9 should also exist at the middle of peak 1 and 2 with height ~67% of peak 10 (Supplementary Note 3). But if considering that the axial microwave also contributes to the EPR signal, the relative peak height

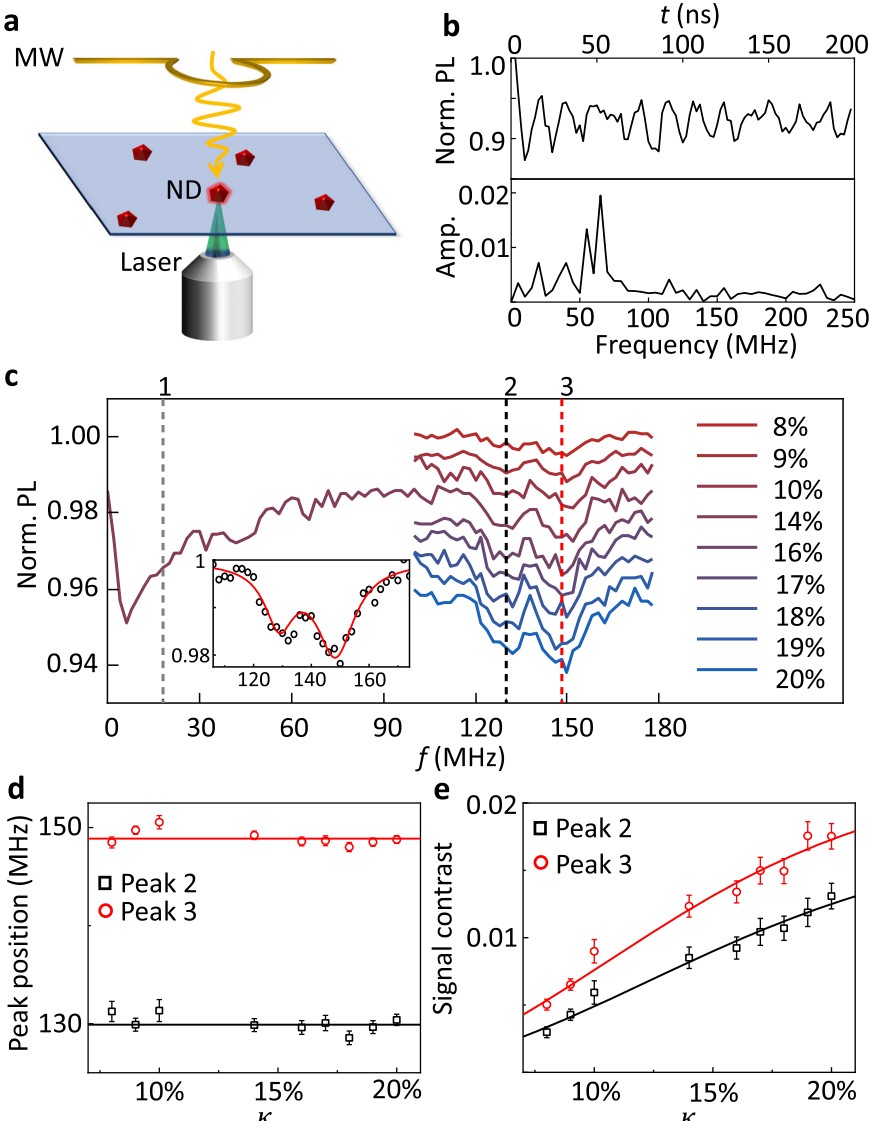

**Fig. 2 | Experimental demonstrations on fixed NDs. a** Sketch of the experimental setup. The NDs are dispersed and fixed on a coverslip, which is placed in a confocal microscope. Yellow wire indicates the coplaner waveguide to radiate microwave. **b** Rabi oscillation. The top is time-domain data, while the bottom is a FFT spectrum. The highest peak indicates $\Omega = 65$ MHz. The input microwave power is 0.6 W. **c** Zero-field EPR spectra of P1 centers with different relative driving index $\kappa$. The three vertical dash lines marked 1, 2, and 3 indicate the theoretical peak positions of

$^{14}$N P1 centers. Inset indicates a two-peak Lorentz fit on a representative spectrum to extract the peak contrasts and positions. **d** Dependence of the peak position on $\kappa$. The points are fitted results, where error bars are fitting errors. The horizontal lines indicate the mean values of 129.9 MHz (peak 2) and 148.9 MHz (peak 3). **e** Dependence of the signal contrast on $\kappa$. The points are fitted results, where error bars are fitting errors. The lines are fits according to Eq. (6).

may reduce down to ~32% (Supplementary Note 5). Such a weak signal is hardly to observe with current signal-to-noise ratio.

The theoretical frequencies of peak 1, 2, and 10 are $4A_\perp$, $\sqrt{A_\parallel^2 + 15A_\perp^2}$, and $\left( \sqrt{A_\parallel^2 + 15A_\perp^2} + \sqrt{4A_\parallel^2 + 12A_\perp^2} \right)/2$, respectively. We then use these values as peak positions to fit the spectrum (Fig. 3d). The fitted result gives $A_\perp^{\text{fit}} = 195 \pm 2$ MHz and $A_\parallel^{\text{fit}} = 579 \pm 8$ MHz, which is slightly different from previous measurements with conventional EPR[33]. Quantitative calculation shows that the signal contrast in Fig. 3d can hardly be explained by freely diffused ions (Supplementary Note 3). There may exist an absorption layer of vanadyl ions on the ND surface[34], which contributes the main signal. Since the hyperfine constants of vanadyl ions strongly depends on the local ligand environment[35], we think the diamond surface might change this environment, and thus lead to different hyperfine interaction.

Repeated measurements on different NDs show different results (Supplementary Note 6), suggesting that the hyperfine constant is indeed ND dependent. Here we cannot perform a blank control because the NV centers in the NDs are slowly losing spin contrast. To speed up the measurement, we repeat the experiment on ND ensembles (Supplementary Note 6), which confirms that the signal is indeed coming from the vanadyl ions. The additional line broadening in the ensemble ND-EPR spectrum is consistent with the assumption of ND-dependent hyperfine constant. Moreover, ensemble measurements on conventional EPR spectrometers rule out the dependence of the hyperfine constant on glycerol ligands (Supplementary Note 7). As the ND contains multiple NV centers, each of them may give a different spectrum due to the different position in the ND. However, considering the signal strongly depends on the NV depth (defined by the minimum distance of the

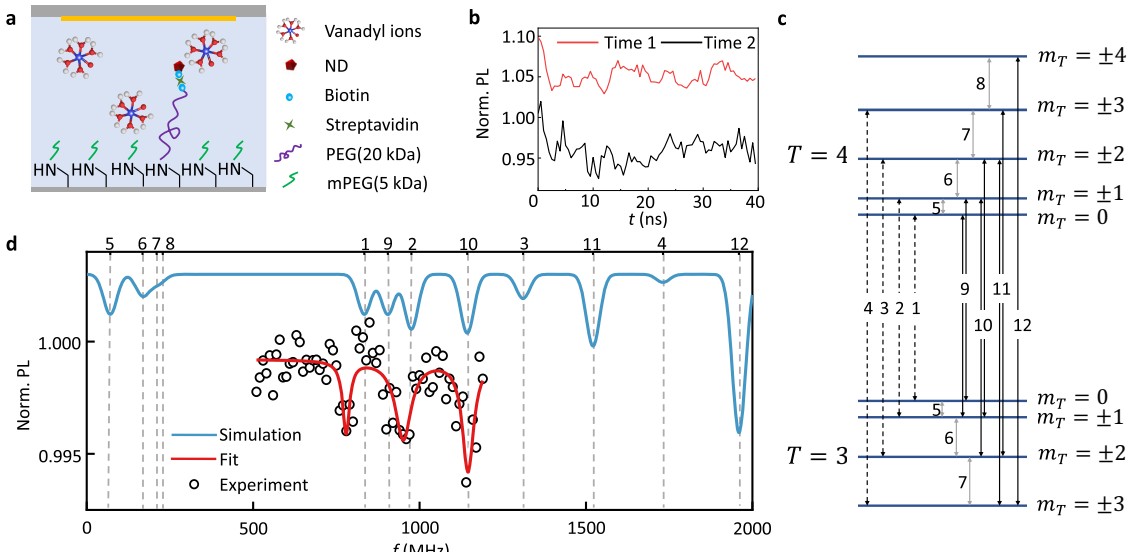

**Fig. 3 | Detection of vanadyl ions with tumbling NDs. a** Feature of the sensor and the target. The ND is tethered by a long polyethylene glycol (PEG) molecule and immersed by a solution of vanadyl sulfate. The short blank PEG is used to control the density of NDs. **b** Rabi oscillation. The two lines are measurements at different times, confirming the existence of a rotational motion of the ND. **c** Energy levels of vanadyl ions. The arrows mark all the allowable transitions. The dash and solid arrows correspond to $\Delta m_T = 0$ and $\pm 1$, respectively. The gray and black solid arrows correspond to $\Delta T = 0$ and 1, respectively. **d** Zero-field EPR spectra of 25 mM vanadyl ions. The blue line is a simulated spectrum with $A_\perp = 208.5$ MHz and $A_\parallel = 547$ MHz. The vertical dash lines mark the peak positions. The points are the experimental result, while the red line is a three-peak Lorentz fit with peak position determined by $A_\perp$ and $A_\parallel$. The fitted linewidth of the three peaks are 25 MHz, 58 MHz, and 42 MHz. The measurement sequence is repeated 16 million times with a duty cycle of 1:19 and a total time consumption of 7 days.

NV center from the ND surface), it is more likely that the shallowest NV will dominate the signal.

According to Eq. (5), the spectral linewidth, defined by the full width at the half maximum (FWHM), is determined by $2\Gamma_2 = 2\Gamma_{2,\text{NV}} + 2\Gamma_{2,\text{V O}}$. For the ND we used, $\Gamma_{2,\text{NV}} \sim 12$ MHz is estimated from the resonance spectrum of the NV center itself (Supplementary Note 2). The relaxation of the vanadyl ion $\Gamma_{2,\text{V O}}$ is contributed by the intrinsic relaxation $R_{\text{VO}}^{\text{int}}$, the dipole-dipole interaction between ions $R_{\text{VO}}^{\text{dip}}$, and the rotational diffusion of ions $R_{\text{VO}}^{\text{rot}}$[36]. The intrinsic relaxation $R_{\text{VO}}^{\text{int}}$ is estimated to be < 12 MHz according to the X band EPR spectrum[37]. The dipole-dipole mediated relaxation rate $R_{\text{VO}}^{\text{dip}} = c \times 272$ MHz·$M^{-1}$[36]. For $c = 25$ mM, this rate is 7 MHz. The rotational diffusion rate is calculated by[38]

$$R_{\text{rot}} = \frac{k_B T}{8\pi r_0^3 \eta} \approx 2 \text{ MHz}, \qquad (9)$$

where $k_B$ is the Boltzmann constant, $T = 293$ K is the temperature, $r_0 \sim 0.37$ nm is the radius of the aqueous vanadyl ion $[\text{VO}(\text{H}_2\text{O})_5]^{2+}$[37], and $\eta = 0.3$ Pa·s is the viscosity of 9:1 glycerol/water mixtures. $R_{\text{rot}}$ will increase to ~600 MHz in pure water, and the measurements will be impossible. Note the transitional diffusion of the vanadyl ion also contributes to the line broadening[36], but is negligible here (~kHz) because the viscosity of the glycerol/water mixtures is much higher than the pure water. Therefore, the estimated linewidth is ≲66 MHz, roughly consisting with the measured spectrum. Here $R_{\text{VO}}^{\text{dip}}$ is underestimated and $R_{\text{rot}}$ is overestimated for ions in the absorption layer. As we perform the measurement at ambient conditions, the Zeeman splitting induced by the geomagnetic field (~50 μT) will also contribute to the line broadening, which is negligible (<2.8 MHz) here.

## Discussion

In conclusion, we have presented a robust method for EPR spectroscopy base on a nanometer-size sensor, even the sensor itself is randomly tumbling. By deploying a amplitude-modulated driving field on the NV center, the resonance condition convert from the NV orientation-dependent driving amplitude to the NV orientation-independent modulation frequency, and thus robust to the tumbling of the host ND. As a demonstration, we show that the zero-field EPR spectrum of P1 centers is indeed robust to the variations of driving amplitude. Moreover, we measure a clear EPR spectrum of vanadyl ions with the ND sensor immersed in a solution of vanadyl sulfate. The extracted hyperfine constants may be used to study the different local environment in the future. This measurement is also robust to the presence of other ions, as the peak positions at zero field are determined solely by the characteristic intrinsic interaction[29,30]. Our method opens the way to nanoscale EPR measurements in complex biological environment, such as in vivo EPR inside a single cell.

The vanadium has been discovered in many biological systems and participates in various biochemical reactions[39], for example, mimic the effect of insulin on glucose oxidation[40], although the mechanism is still unclear. Nanoscale EPR studies of the vanadyl ion may benefit the understanding of its interaction with biological molecules, if the spectral resolution can be improved. Except for the intrinsic relaxation $R_{\text{VO}}^{\text{int}}$, all other components contributing to the line broadening can be removed by some technical improvements. For example, $\Gamma_{2,\text{NV}}$ can be reduced to submegahertz by using high-purity NDs[41]. $R_{\text{VO}}^{\text{dip}}$ can be reduced directly by lowing the ion concentration. $R_{\text{rot}}$ can be removed by measuring the solid-state spectrum. Moreover, even $R_{\text{VO}}^{\text{int}}$ itself can be reduced to ~2 MHz if utilizing the noise-insensitive transitions[42]. We note that the fundamental limit maybe even better than this value if single ions can be detected[43]. By then, magnetic shielding or compensation will be required.

Although we have solved the problem of random tumbling of NDs, there are still other challenges for biological applications of ND-EPR, such as cellular uptake of NDs, microwave heating, and measurement efficiency. The ND we used may be too large (~40 nm) to be compatible with single-cell studies. Fortunately, some progress has been made in reducing the ND size[41,44,45]. It has been reported that even 5-nm NDs can contain NV$^-$ centers[46]. Reduction of the ND size will inevitably lead to poorer charge-state stability and coherence time of

NV centers within the ND. However, for near-surface NV centers with the same depth, the ND size will play a marginal role. Since our scheme is highly surface-sensitive (Supplementary Note 3), smaller NDs will not deteriorate the performance, but increase the conditional probability of finding near-surface NV centers. Besides, surface engineering of NDs is another effective way to improve cellular uptake[47]. It is also helpful in increasing the charge-state stability and the coherence time of near-surface NV centers[48,49].

The heating effect is the main damage caused by microwave radiation to living cells[50]. It is indeed a problem for the detection of vanadyl ions and other paramagnetic targets with strong intrinsic interactions, because the resonance frequency and accordingly required microwave power are high. A direct but inefficient way to control the average microwave power is to prolong the idle time. During our measurement of vanadyl ions, the duty cycle is 1:19, corresponding to an average power of <1 W. A better way is to optimize the microwave antenna to improve radiation efficiency. In the future, our demonstration can be generalized to the detection of radicals with a relatively lower resonance frequency[17,30], and then the issue of microwave heating can be directly avoided.

The current demonstration is still time consuming due to the poor measurement efficiency and signal contrast. For instance, the data in Fig. 3d takes almost a week with 95% of the time wastes controlling the average microwave power. The spin-to-charge conversion technique[51] may be an excellent solution, because it can not only achieve better readout fidelity[52,53], but also utilize the idle time to perform the slow charge-state readout. As described above, surface engineering is a possible way to improve the property of near-surface NV centers, which can thus increase the signal contrast. For example, the current Rabi contrast of only 10% (Fig. 3b) can be improved to ~30% via the charge-state improvement, corresponding to a signal enhancement of 3 and time saving of nearly an order of magnitude. A further advantage is that shallower NV centers may be usable, which will significantly increase the signal contrast due to its strong dependence on the NV depth. Considering the spectrum is robust to the orientation of both the sensor and the target, we can utilize an ensemble of NDs simultaneously to achieve high parallel efficiency, although with a loss of spatial resolution.

## Methods

### Experimental setup
The optical part of our setup is a home-built confocal microscopy, where a diode laser (CNI MGL-III-532) generates the excitation light, and the photoluminescence is detected by an avalanche photodiode (Perkin Elmer SPCM-AQRH-14). The microwave part consists of an arbitrary waveform generator (Keysight M8190a), a microwave amplifier (Mini-circuits ZHL-16W-43+), and a coplanar waveguide.

### Chemical preparations of tumbling NDs
The surface of the ND we use (Adámas, Carboxylated 40 nm Red Fluorescent ND in DI water, ≤1.5 ppm NV, 12–14 NV centers per particle, 1 mg/mL) is originally terminated with carboxyl groups. To realize the biotinylation, we cover the ND surface with amine-PEG3-biotin. The detailed procedure is as follows: we freshly prepare 100 uL solution containing 1mM amine-PEG3-biotin (EZ-Link™), 5 mM EDC (1-[3-(Dimethylamino)propyl]-3-ethylcarbodiimide methiodide, Sigma-Aldrich) in 100 mM MES (4-Morpholineethanesulfonic acid sodium salt, pH 5.0), and mix it with 10 μL ND suspension. The reaction is allowed to proceed at room temperature for 30 min. We then add 20 μL of 5 mM EDC to the ND mixture and wait 30 min, repeating two times to maximize the amount of amine-PEG3-biotin bound to the ND surface.

A coverslip is used as the substrate for bonding the NDs. Before use, the coverslip is thoroughly cleaned by the following procedure.

First, we sonicate the coverslip with MilliQ water for 15 min to remove dirt. After that, we replace the MilliQ water with acetone, and sonicate the coverslip for a further 15 min, rinsing it three times with MilliQ water to remove any acetone residue. The coverslip is then sonicated with 1 M KOH for 20 min and rinsed with MilliQ water. Finally, we immerse the coverslip in Piranha solution (3:1 mixture of concentrated sulfuric acid and 30% hydrogen peroxide) for 30 min at 90 °C and rinse it with MilliQ water. After cleaning, we modify the surface of the coverslip with amino group. We prepare the aminosilylation solution by adding 10 mL of methanol, 0.5 mL of acetic acid, and 0.3 mL of APTES (3-aminopropyltrimethoxysilane, Sigma-Aldrich) to a clean beaker. Then we rinse the cleaned coverslip with methanol and sink it in the aminosilylation solution. The reaction is allowed to proceed at room temperature for 20–30 min, during which time the coverslip is sonicated in the aminosilylation solution once for 1 min. The coverslip is then rinsed three times with methanol. To tether the NDs to the surface of the coverslip and to maintain the rotational movement of the NDs, we bind long-chain biotinylated PEG to the surface of the coverslip, and use short-chain mPEG to control the density of the biotin termination. We prepare a PEG mixture of 0.8 mg biotinylated NHS-ester PEG (20,000 Da, Aladdin) and 8 mg of NHS-ester mPEG (5000 Da, Aladdin) in a 100 μL tube, add 64 μL of 0.1 M NaHCO₃ solution, and pipette it to dissolve them completely. We then drop the PEGylation solution onto the amino-silanated coverslip and keep the environment moist to prevent the solution from drying out. We incubate the coverslip in a dark and humid environment for 5 h, then rinse the coverslip with MilliQ water.

The final step is to attach the biotinylated ND to the biotinylated coverslip using streptavidin. We drop 1 mg/mL streptavidin solution (Sangong Biotech) onto the biotinylated coverslip, wait for 30 min and then rinse the coverslip with MilliQ water. Then we add the biotinylated ND suspension obtained in the previous step to the coverslip, wait for 30 min again and rinse the coverslip with MilliQ water.

### Chemical preparations of vanadyl ions
We dissolve the vanadium sulfate pentahydrate powder in the deoxygenated MilliQ water to make a 100 μL 250 mM $VO^{2+}$ solution, then mix it with 900 μL deoxygenated glycerol to obtain glycerol aqueous solution (glycerol:water = 9:1) of 25 mM $VO^{2+}$. The solvents are deoxygenated to prevent oxidation of the vanadyl ions. The detailed deoxygenation operations are purging the MilliQ water with $N_2$ under reduced pressure and placing the glycerol container in liquid nitrogen, then purging the glycerol with $N_2$ under reduced pressure. To keep the solution acidic, we add 10 μL of 1 M sulfuric acid prepared with deoxygenated MilliQ water to the solution and mix it thoroughly. We then seal a tiny drop of the solution between the ND-bonded coverslip and the coplanar waveguide. All the above operations are done under nitrogen atmosphere in a glove box.

## Data availability
Data supporting the findings of this study are available within the article and its Supplementary Information file.

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

## Acknowledgements

We thank X. C. Su and S. Wang for helpful discussions. This work was supported by the National Natural Science Foundation of China (Grant Nos. T2125011 to F.S., 31971156 to F.K., 81788101 to J.D.), the National Key R&D Program of China (Grant Nos. 2021YFB3202800 to S.C., 2018YFA0306600 to J.D.), the CAS (Grant Nos. XDC07000000 to J.D., GJJSTD20200001 to J.D., Y201984 to F.S., YSBR-068 to F.S.), Innovation Program for Quantum Science and Technology (Grant Nos. 2021ZD0302200 to J.D., 2021ZD0303204 to F.S.), the Anhui Initiative in Quantum Information Technologies (Grant No. AHY050000 to J.D.), Hefei Comprehensive National Science Center, and the Fundamental Research Funds for the Central Universities. This work was partially carried out at the USTC Center for Micro and Nanoscale Research and Fabrication.

## Author contributions

J.D. and F.S. supervised the entire project. F.K. and F.S. designed the experiments. Z.Q., J.S., S.C., and Q.Z. prepared the sample. Z.Q., Z.W., F.K., and P.Z. performed the experiments. F.K. and Z.H. carried out the calculations. F.K., Z.Q., and F.S. wrote the manuscript. All authors discussed the results and commented on the manuscript.

## Competing interests

The authors declare no competing interests.
