## [Peer Review File · Nature Communications]

REVIEWER COMMENTS

Reviewer #1 (Remarks to the Author):

This manuscript proposed a protocol to perform zero-field EPR with nanodiamonds. The authors proposed an amplitude modulation sequence to overcome the challenge of random NV orientation in the nanoparticles. They demonstrated EPR spectrum of P1 centers in diamonds measured by such sensors. And they also conducted a proof-of-concept measurement of vanadyl ions. In general, I find the study to be solid. If the authors are able to do some major improvements and clarifications, I would consider it to be qualified to publish at Nature Communications.

1. The authors claimed to use this nanoscale EPR sensor to do biological studies. There are some challenges need to be addressed or clarified:

a. The particles being used are large (~40nm), and this may not be compatible to single-cell level studies. And it is not trivial to use smaller size of particles, because the surface quality of the smaller particles will be largely deteriorated, and the charge state stability of the defect centers and the coherence time will be affected significantly. Moreover, the crystallinity of the diamond will not preserve well in smaller particles.

b. The linewidth of the spectrum is ~56MHz. The authors should comment on the fundamental limit of this number. This large linewidth will already limit the application of this approach. The authors should characterize how the vanadium EPR performs in conditions when other ions present.

c. The SNR of the approach is poor. Eg. the SNR of FIG 3d is low. And it is not obvious that the challenge can be resolved by reducing the particle sizes due to the reasons above.

d. The authors should comment on the "Specific Absorption Rate" or "SAR" of the approach.

2. The authors should clarify the conditions where the approach can work.

a. The authors claimed that this is a zero-field approach, and the manuscript stated that "in the absence of external field". What is the exact magnetic field strength and if any shielding has been applied to achieve this should be clarified.

b. It seems like $|+1\rangle$ and $|-1\rangle$ states for NV in FIG1 is degenerate, but there is a field that causes splitting on the target spin in Equation (1). The authors should clarify this.

c. In general, the authors should comment on the limitation of this approach and the field conditions where it will work. It will also be helpful to comment on its performance under noisy conditions.

3. FIG 2 demonstrated a P1 EPR spectrum when varying relative driving index k , but there are some aspects need to be clarified. The authors should show some experiments with nanodiamonds that are oriented differently. I think the claim that different k simulates different orientation is not so evident.

4. The authors should compare their approach with other nanodiamond sensing techniques such as, Igarashi, Ryuji, et al. "Tracking the 3D rotational dynamics in nanoscopic biological systems." *Journal of the American Chemical Society* 142.16 (2020): 7542-7554.

Reviewer #2 (Remarks to the Author):

This manuscript reports on an interesting new scheme for detecting zero-field EPR spectra with a single nanodiamond (ND). The experiment is demonstrated on a (probably) highly concentrated solution of vanadyl ions in a high-viscosity solvent. This is clearly high-level work of broad interest. Yet, its presentation leaves open a number of important question that affect interpretation of the experimental results and assessment of the application potential of the technique. Therefore, I recommend that decision on publication in *Nature Communications* is made only after a revision that addresses the following points.

1. Page 3, left column, line 4 from bottom: "Since the signal strength depends on the relative driving index κ (Eq. 5)" I do not understand the reference to Eq. (5) here. I would have written " κ proportional to $B1/f$ " here. Strictly speaking, $B1$ is the driving amplitude, not the driving power.

2. Please specify the concentration of the vanadyl sulfate solution in the main text and in the caption of Figure 3. Please also specify the total measurement time in the figure caption.

3. The Experimental Section of the main text specifies vanadyl concentration as 25 micromolar. If so, why did you use such a low concentration? This should be explained. Figure S4b specifies the vanadyl concentration for ensemble measurements as 25 millimolar. This in turn would be very high. In the main text, above Eq. (9), you also specify 25 mM. What exactly is the concentration in the two sets of experiments (main text and SI)? A concentration of 25 mM is orders of magnitude away from in-cell concentrations of species of interest.

4. Why do you use glycerol:water 9:1? Is the high viscosity of 0.3 Pa·s a requirement for the experiment to succeed? This is 300 times the viscosity of water (but less than the viscosity than cytoplasm). Please comment.

5. Glycerol is a chelating ligand. It is unlikely that the main species in this solution are $[\text{VO}(\text{H}_2\text{O})_5]^{2+}$ complexes. I would expect that the vanadyl ions are coordinated by either glycerol only or by glycerol and water ligands. This may explain why you measure hyperfine couplings that differ from expectations for $[\text{VO}(\text{H}_2\text{O})_5]^{2+}$.

6. Page 4, right column, line 2: "However, Peaks 5-8 will mix with the strong signal of P1 centers". Many readers will interpret mixing as something caused by spin-spin interaction. What you mean is probably superposition of the peaks caused by similar frequencies. Please write "superimpose" or "overlap" instead of "mix".

7. I am not sure that I understand the argument of 64 time higher sensitivity when reducing the size of the ND by a factor of two. This would imply that your signal is dominated by vanadyl ions close to the ND surface. Sensitivity would then depend very strongly on the distance of the utilized NV centre from the ND surface. As a consequence, I would expect different NDs to provide strongly different sensitivity (however, see also 9.). Did you observe this?

8. Your arguments about differences in hyperfine couplings to bulk measurements as well as between different NDs imply that the signal is dominated by vanadyl ions in contact with the ND surface. Unless you used NDs with NV centres that were fortuitously close to the surface, this would appear unlikely. Assume that the NV centre was 2 nm from the surface (pretty close already for a 40 nm particle). A vanadyl complex in contact with the surface is about 0.2 nm from the surface. At about 0.8 nm I would expect that interaction with the surface is negligible. Signal strength would still be about 45%, but there are more sites 0.8 nm from the NV centre than 0.4 nm from the NV centre. The argument extends to even longer distances. Please state explicitly whether you assume the detected vanadyl ions to be in contact with the ND surface.

9. If you want to uphold the claim that reducing the size of NDs by a factor of two increases signal strength by a factor of 64, you should discuss this in more detail in the Supplementary Material. Such discussion should take into account the (average) location of NV centres with respect to the ND surface and the sensitive volume (r^{-6} weighted outside the ND).

10. You use 40 nm NDs with < 1.5 ppm NV centres. This still means that each ND is likely to feature several (of the order of 20) NV centres. If you agree with this assessment, please mention and discuss this feature of the experiment.

11. Did you observe oxidation of vanadyl ions when you did not deoxygenate the solutions? I would be surprised. Vanadyl ions are expected to be more stable than V(V) species under ambient conditions. The point is of interest, because for in-cell measurements, you would like to work under physiological conditions.

12. The main point of your manuscript is the amplitude-modulated scheme that I also consider as elegant. However, in Supplementary Note 4 you argue that, for the case at hand, the “off-resonance” method is as good. You used the off-resonance method for Supplementary Note 6. You should openly address this point in the main text and you should consider whether there exist cases for which the amplitude-modulated scheme would be substantially superior.

13. You cannot perform a control experiment without vanadyl ions with the same ND, because the measurements are performed for such a long time that the NV centres in the NDs bleach. Instead, you performed such a control experiment on an ND ensemble (Supplementary Note 6). You need to mention this in the main text. Still I am curious why you did not perform control experiments with single NDs in the absence of vanadyl ions. This would have been an obvious control.

14. Figure S4 shows signals at about 2870 MHz. Are they another driving-field artefact (compare Figure S3b)? Something needs to be said about these signals.

Typos:

Page 2, left column, second line from bottom: “by bring the NV center” should read “by bringing the NV center”

Page 3, right column, last line “predication” should read “prediction”

Ref. [4]: ‘structurecreactivity’ should read ‘structure-reactivity’

Reviewer #1 :

This manuscript proposed a protocol to perform zero-field EPR with nanodiamonds. The authors proposed an amplitude modulation sequence to overcome the challenge of random NV orientation in the nanoparticles. They demonstrated EPR spectrum of P1 centers in diamonds measured by such sensors. And they also conducted a proof-of-concept measurement of vanadyl ions. In general, I find the study to be solid. If the authors are able to do some major improvements and clarifications, I would consider it to be qualified to publish at Nature Communications.

Reply:

We thank the reviewer for the concise summary. We have carefully revised the manuscript according the reviewer's comments.

1. The authors claimed to use this nanoscale EPR sensor to do biological studies. There are some challenges need to be addressed or clarified:
 - a. The particles being used are large (~40nm), and this may not be compatible to single-cell level studies. And it is not trivial to use smaller size of particles, because the surface quality of the smaller particles will be largely deteriorated, and the charge state stability of the defect centers and the coherence time will be affected significantly. Moreover, the crystallinity of the diamond will not preserve well in smaller particles.

Reply:

We thank the reviewer for pointing out the insufficient discussion. We have clarified the challenges of biological applications and discussed possible solutions for them in the revised manuscript (third paragraph in the Discussion section). In short, reducing the size of nanodiamond is a direct way to improve the compatibility in cells. And it will indeed deteriorate the coherence time and the charge-state stability of NV centers inside the nanodiamond. The reason is the smaller size, the shorter average NV-surface distance. But for two specific NV centers with the same NV-surface distance, as shown in the figure below, their spin and charge properties will be nearly the same.

Considering that only shallow NV centers contribute to the EPR signal (see the revised supplementary note 3), the performance of our scheme on smaller nanodiamonds will not be affected. Another way to improve the cell compatibility of large particles is surface

engineering [Bioconjugate Chem. 2018, 29, 8, 2786–2792]. It can also improve the surface quality [ACS Appl. Mater. Interfaces 10, 13143–13149 (2018); Nanoscale 11, 1770–1783 (2019)]. We are also working on this area, and continuously pushing this nanoscale EPR technique to practical biological applications.

b. The linewidth of the spectrum is ~56MHz. The authors should comment on the fundamental limit of this number. This large linewidth will already limit the application of this approach. The authors should characterize how the vanadium EPR performs in conditions when other ions present.

Reply:

We have added a detailed discussion on the linewidth in the revised manuscript (second paragraph in the Discussion section). The fundamental limit is ~20 MHz for normal transitions, and ~2 MHz for noise-insensitive transitions. If one can detect a single ion in the future, this number may be even smaller. A key figure of zero-field EPR is the resonance frequencies are determined solely by the intrinsic interaction, and independent on their orientations. Different ions will have different characteristic spectra [Chem. Rev. 83, 49–82 (1983)]. So, the zero-field vanadium EPR spectrum is robust to the presence of other ions. In a previous work [Nat. Commun. 9, 1563 (2018)], we have already shown that the target spin can be resolved from bath spins by zero-field EPR technique. We have added a brief description about this point in the revised manuscript (first paragraph in the Discussion section).

c. The SNR of the approach is poor. Eg. the SNR of FIG 3d is low. And it is not obvious that the challenge can be resolved by reducing the particle sizes due to the reasons above.

Reply:

We admit that the SNR of the current proof-of-principle demonstration is not satisfactory. We have added a detailed discussion in the revised manuscript (last paragraph in the Discussion section) to clarify how to improve the SNR or measurement efficiency. Indeed, reducing the ND size is not a direct solution. Possible improvements include utilizing spin-to-charge conversion technique to improve the readout efficiency, surface engineering to improve signal contrast, replacing the target with radicals to save the idle time.

d. The authors should comment on the "Specific Absorption Rate" or "SAR" of the approach.

Reply:

We think the SAR mentioned by the reviewer refer to the radiation damage of microwave. The main effect of microwave radiation is the heating effect [Phys. Rev. Applied 13, 024021 (2020)]. For biological applications, the average microwave power is limited. So, one can either reduce the driving power, or increase the idle time. We have added a detailed discussion about this issue in the revised manuscript (fourth paragraph in the Discussion section).

2. The authors should clarify the conditions where the approach can work.

a. The authors claimed that this is a zero-field approach, and the manuscript stated that "in

the absence of external field". What is the exact magnetic field strength and if any shielding has been applied to achieve this should be clarified.

Reply:

We perform the experiment at ambient conditions without any shielding. We have clarified it in the revised manuscript (last paragraph in the section "EPR measurement with tumbling NDs").

b. It seems like $|+1\rangle$ and $|-1\rangle$ states for NV in FIG1 is degenerate, but there is a field that causes splitting on the target spin in Equation (1). The authors should clarify this.

Reply:

The energy splitting of the target spin is induced by the intrinsic interaction, such as hyperfine interaction. We have clarified it in the revised manuscript (sentences below Eq. (1)). We give a detailed calculation of this splitting in supplementary note 3.

c. In general, the authors should comment on the limitation of this approach and the field conditions where it will work. It will also be helpful to comment on its performance under noisy conditions.

Reply:

The vanadium EPR can be performed at ambient condition at the present stage, due to the line broadening induced by the geomagnetic field is negligible. But if pushing the linewidth to the fundamental limit, magnetic shielding or compensation will be required, as discussed in [Sci. Adv. 6, eaaz8244 (2020)]. We have clarified it in the revised manuscript. We are not clear about the noisy conditions mentioned by the reviewer. Our measurement is already performed in a noisy condition, which manifests in the strong relaxations of both the sensor and the target. The dependence of the EPR spectrum on noise is given by Eq. 5, Eq. 6, and also Eq. S11. If the noisy condition means the presence of other paramagnetic ions, we have addressed this issue above (Reply to comment 1b, and first paragraph in the Discussion section).

3. FIG 2 demonstrated a P1 EPR spectrum when varying relative driving index k , but there are some aspects need to be clarified. The authors should show some experiments with nanodiamonds that are oriented differently. I think the claim that different k simulates different orientation is not so evident.

Reply:

Previous measurements depend on the orientation of nanodiamonds, because the effective driving strength depends on the orientations. So, we show the robustness on driving strength to simulate the robustness on orientation. To confirm that, we have performed additional measurements on multiple NDs with random orientations, as given in the revised manuscript (supplementary note 6).

4. The authors should compare their approach with other nanodiamond sensing techniques such as, Igarashi, Ryuji, et al. "Tracking the 3D rotational dynamics in nanoscopic biological systems." *Journal of the American Chemical Society* 142.16 (2020): 7542-7554.

Reply:

Thank the reviewer for pointing out this interesting work. It is an active way to deal with the tumbling of nanodiamonds, while our scheme is a passive way. We have added a comparison in the revised manuscript (introduction section).

Reviewer #2:

This manuscript reports on an interesting new scheme for detecting zero-field EPR spectra with a single nanodiamond (ND). The experiment is demonstrated on a (probably) highly concentrated solution of vanadyl ions in a high-viscosity solvent. This is clearly high-level work of broad interest. Yet, its presentation leaves open a number of important questions that affect interpretation of the experimental results and assessment of the application potential of the technique. Therefore, I recommend that decision on publication in Nature Communications is made only after a revision that addresses the following points.

Reply:

We appreciate that the reviewer recommends our work for publication in Nature Communications. As suggested by the reviewer, we have carefully revised the manuscript.

1. Page 3, left column, line 4 from bottom: "Since the signal strength depends on the relative driving index κ (Eq. 5)" I do not understand the reference to Eq. (5) here. I would have written " κ proportional to B_1/f " here. Strictly speaking, B_1 is the driving amplitude, not the driving power.

Reply:

Thanks for the suggestion. We have corrected it accordingly.

2. Please specify the concentration of the vanadyl sulfate solution in the main text and in the caption of Figure 3. Please also specify the total measurement time in the figure caption.

Reply:

Thanks for the suggestion. We have corrected it accordingly.

3. The Experimental Section of the main text specifies vanadyl concentration as 25 micromolar. If so, why did you use such a low concentration? This should be explained. Figure S4b specifies the vanadyl concentration for ensemble measurements as 25 millimolar. This in turn would be very high. In the main text, above Eq. (9), you also specify 25 mM. What exactly is the concentration in the two sets of experiments (main text and SI)? A concentration of 25 mM is orders of magnitude away from in-cell concentrations of species of interest.

Reply:

We apologize for the typos. The ion concentration for all the measurements is 25 mM. We have corrected it in the revised manuscript. The current measurement is a proof-of-principle demonstration. With proper improvements, such as chemical modifications of ND surface to capture the target ions, the detectable concentration can be much lower.

4. Why do you use glycerol:water 9:1? Is the high viscosity of 0.3 Pa·s a requirement for the

experiment to succeed? This is 300 times the viscosity of water (but less than the viscosity than cytoplasm). Please comment.

Reply:

We used a high-viscosity solvent because the rotational diffusion rate R_{rot} is inversely proportional to the viscosity η (Eq. 9). For 9:1 glycerol aqueous solution, R_{rot} is 2 MHz. For pure water, R_{rot} is 600 MHz. We have added some clarification in the revised manuscript.

5. Glycerol is a chelating ligand. It is unlikely that the main species in this solution are $[\text{VO}(\text{H}_2\text{O})_5]^{2+}$ complexes. I would expect that the vanadyl ions are coordinated by either glycerol only or by glycerol and water ligands. This may explain why you measure hyperfine couplings that differ from expectations for $[\text{VO}(\text{H}_2\text{O})_5]^{2+}$.

Reply:

Thanks for the suggestion. To confirm the reviewer's expectation, we have performed ensemble measurements on conventional EPR spectrometers. The result is shown in the revised manuscript (supplementary note 7). In short, the liquid EPR spectrum in 1:3 glycerol aqueous solution is the same with that in pure water, which means low concentration of glycerol will not change the hyperfine coupling of vanadyl ions. The low-temperature powder EPR spectrum in 1:3 glycerol aqueous solution is the same with that in 9:1 glycerol aqueous solution, which means high concentration of glycerol will also not change the hyperfine coupling of vanadyl ions. Therefore, the glycerol ligand seems not coordinate with the vanadyl ion, or at least not change the hyperfine constant. We did not make a direct comparison between the 9:1 glycerol aqueous solution and the pure water, because we can neither acquire liquid EPR spectra in such a viscous solution, nor acquire powder EPR spectra in pure water.

6. Page 4, right column, line 2: "However, Peaks 5-8 will mix with the strong signal of P1 centers". Many readers will interpret mixing as something caused by spin-spin interaction. What you mean is probably superposition of the peaks caused by similar frequencies. Please write "superimpose" or "overlap" instead of "mix".

Reply:

Thanks for the suggestion. We have corrected it in the revised manuscript.

7. I am not sure that I understand the argument of 64 time higher sensitivity when reducing the size of the ND by a factor of two. This would imply that your signal is dominated by vanadyl ions close to the ND surface. Sensitivity would then depend very strongly on the distance of the utilized NV centre from the ND surface. As a consequence, I would expect different NDs to provide strongly different sensitivity (however, see also 9.). Did you observe this?

Reply:

Sorry for the misleading argument. We have added a detailed discussion about the signal in

the revised manuscript (supplementary note 3 and discussion section in the main text). The EPR signal do not directly depend on the ND size, but indeed strongly depends on the NV-surface distance. We did observe that the signal strength on different NDs varies greatly.

8. Your arguments about differences in hyperfine couplings to bulk measurements as well as between different NDs imply that the signal is dominated by vanadyl ions in contact with the ND surface. Unless you used NDs with NV centres that were fortuitously close to the surface, this would appear unlikely. Assume that the NV centre was 2 nm from the surface (pretty close already for a 40 nm particle). A vanadyl complex in contact with the surface is about 0.2 nm from the surface. At about 0.8 nm I would expect that interaction with the surface is negligible. Signal strength would still be about 45%, but there are more sites 0.8 nm from the NV centre than 0.4 nm from the NV centre. The argument extends to even longer distances. Please state explicitly whether you assume the detected vanadyl ions to be in contact with the ND surface.

Reply:

We have added a detailed calculation in the revised supplement note 3. As the signal strongly depends on the NV depth h , only shallow NV centers contribute to the detected signal. For freely diffused ions, the signal decays as $\sim h^{-3}$ (Eq. S31). Following the reviewer's assumption, the ions >0.8 nm from the surface will contribute about 70% of the total signal. From this point of view, it can hardly to see the signal is dominated by ions in contact with the surface. However, the quantitative calculation shows that the NV depth should be ~ 1 nm, if the signal in Fig. 3d comes from freely diffused ions with concentration of 25 mM. Existence of such a shallow NV center seems impossible. So, it is more likely the ions in contact with ND surface have a higher local concentration, such as staying in an adsorption layer. Considering the glycerol ligand can not explain the difference in hyperfine coupling, a better explanation is the measured EPR signal is dominated by ions absorbed on the ND surface.

9. If you want to uphold the claim that reducing the size of NDs by a factor of two increases signal strength by a factor of 64, you should discuss this in more detail in the Supplementary Material. Such discussion should take into account the (average) location of NV centres with respect to the ND surface and the sensitive volume (r^{-6} weighted outside the ND).

Reply:

Thanks for the suggestion. We have added a detailed discussion accordingly.

10. You use 40 nm NDs with < 1.5 ppm NV centres. This still means that each ND is likely to feature several (of the order of 20) NV centres. If you agree with this assessment, please mention and discuss this feature of the experiment.

Reply:

The data provided by the company is 12-14 color centers per particle. We have clarified it in the revised manuscript (section "EPR measurements with fixed NDs", "EPR measurements with tumbling NDs", and "Chemical preparations of tumbling NDs").

11. Did you observe oxidation of vanadyl ions when you did not deoxygenate the solutions? I would be surprised. Vanadyl ions are expected to be more stable than V(V) species under ambient conditions. The point is of interest, because for in-cell measurements, you would like to work under physiological conditions.

Reply:

The vanadyl ions is quite stable. But we still observed that the color of the solution of vanadyl sulfate will change slowly (several weeks) under ambient conditions. Considering that the current measurement is time consuming, we deoxygenate the solution just for insurance.

12. The main point of your manuscript is the amplitude-modulated scheme that I also consider as elegant. However, in Supplementary Note 4 you argue that, for the case at hand, the "off-resonance" method is as good. You used the off-resonance method for Supplementary Note 6. You should openly address this point in the main text and you should consider whether there exist cases for which the amplitude-modulated scheme would be substantially superior.

Reply:

Thanks for the suggestion. We have made a short comparison of the two schemes in the revised main text. We also give a figure (Fig. S4 in the revised supplementary note 4) to clearly show the difference between the two schemes. In short, they are similar when the spectral resolution is poor, and the amplitude-modulated scheme will outperform when the spectral resolution is improved.

13. You cannot perform a control experiment without vanadyl ions with the same ND, because the measurements are performed for such a long time that the NV centres in the NDs bleach. Instead, you performed such a control experiment on an ND ensemble (Supplementary Note 6). You need to mention this in the main text. Still I am curious why you did not perform control experiments with single NDs in the absence of vanadyl ions. This would have been an obvious control.

Reply:

Thanks for the suggestion. We have mentioned it in the revised main text. We did not perform blank control on single NDs because the signal strength on different NDs varies greatly, as described above. Therefore, it is hard to tell whether the disappearance of signal comes from the absence of vanadyl ions or the use of "bad" NDs.

14. Figure S4 shows signals at about 2870 MHz. Are they another driving-field artefact (compare Figure S3b)? Something needs to be said about these signals.

Reply:

We apologize for the misleading figure. To clarify, we have added the corresponding pulse sequences in the revised figure (now Fig. S7). Signals appear at 2870 MHz because they are

normal optically detected magnetic resonance (ODMR) spectra of the NV centers itself at zero magnetic field.

Typos:

Page 2, left column, second line from bottom: “by bring the NV center” should read “by bringing the NV center”

Page 3, right column, last line “predication” should read “prediction”

Ref. [4]: ‘structurecreactivity’ should read ‘structure-reactivity’

Reply:

Sorry for the typos. We have made corrections accordingly and carefully polished the language throughout.

REVIEWERS' COMMENTS

Reviewer #1 (Remarks to the Author):

The authors addressed most of the comments and clarified most of the ambiguous statements I mention in the initial reviews.

Still I would like to comment on the following aspects.

- Regarding the response to the coherence deterioration with smaller samples (#1), if you care about only the NVs that have same distance to the surface, considering the number of these defects you are concerned, it will reduce with r^2 , where r is the size of the particle. And the SNR will reduce significantly. Also the crystallinity concern is not addressed. The crystal structure in small sizes will never be perfect. I in general suggest not to advocate biological applications given what the authors have shown.

- The authors took the convention "zero-field" in a wrong way. Please refer to "zero-field" NMR papers, such as Weitekamp, D. P., et al. "Zero-field nuclear magnetic resonance." Physical review letters 50.22 (1983): 1807. Ambient field is not zero field.

Overall, I still give my positive perspective for this manuscript. But the authors should be careful to not overclaim.

Reviewer #2 (Remarks to the Author):

I am satisfied with the rebuttal and with the changes made in the manuscript in Supplementary Notes. The work is now presented in a balanced way and some previously hard-to-understand points have been clarified. The English of the new text sections is not quite on the same level as the English in the original manuscript. As this does not lead to ambiguity, technical editors of Nat. Comm. will probably fix it, but you should carefully check your proofs. Nowadays, you do not need a native English speaker for writing reasonable and correct English. Just translate your English text to Chinese by DeepL (or similar software), check that the Chinese version is fine, and translate it back to English.

Reviewer #1 :

The authors addressed most of the comments and clarified most of the ambiguous statements I mention in the initial reviews.

Still I would like to comment on the following aspects.

- Regarding the response to the coherence deterioration with smaller samples (#1), if you care about only the NVs that have same distance to the surface, considering the number of these defects you are concerned, it will reduce with r^2 , where r is the size of the particle. And the SNR will reduce significantly. Also the crystallinity concern is not addressed. The crystal structure in small sizes will never be perfect. I in general suggest not to advocate biological applications given what the authors have shown.

Reply:

The number of near-surface NV centers indeed reduces with r^2 . However, the number of total NV centers reduces with r^3 . The conditional probability of finding a near-surface NV center, given that the nanodiamond contains NV centers, is proportional to $1/r$. We note that the selection of nanodiamonds containing NV centers is trivial (via fluorescence and ODMR measurements), so our EPR measurement depends on the conditional probability ($1/r$) rather than r^2 . We have clarified this in the revised manuscript.

The fabrication of high-profile small nanodiamond is an active research topic (ref. 41,44-46 in the main text). The crystallinity is not a fatal problem. For instance, nanodiamonds as small as 5 nm can contain NV centers (D. Terada et al, ACS nano (2019)), although the crystal structure maybe imperfect. We agree with the reviewer that biological applications are challenging, but the difficulties can be solved step by step. We have taken a step forward and will continue to do so.

- The authors took the convention "zero-field" in a wrong way. Please refer to "zero-field" NMR papers, such as Weitekamp, D. P., et al. "Zero-field nuclear magnetic resonance." Physical review letters 50.22 (1983): 1807. Ambient field is not zero field.

Reply:

Thanks for the reminder. In this work, we focus on EPR rather NMR. Actually, the zero-field EPR has a longer history (G. S. Bogle, et al Proc. Phys. Soc. 77 561 (1961)). The need to distinguish between ambient and zero fields depends on whether there are observable differences in the corresponding spectra. For zero-field NMR, the intrinsic interactions are on the order of 100 Hz with a high spectral resolution (\sim mHz). Even an ultralow magnetic field (\sim 100 nT) can change the NMR spectrum dramatically. Therefore, careful magnetic shielding is usually

required. However, for the EPR measurements in this work, the intrinsic interactions are much higher (~ 1000 MHz) with a spectral resolution of ~ 10 MHz. The ambient field (~ 50 uT) will not change the EPR spectrum obviously, as we discussed in the main text.

Overall, I still give my positive perspective for this manuscript. But the authors should be careful to not overclaim.

Reviewer #2 :

I am satisfied with the rebuttal and with the changes made in the manuscript in Supplementary Notes. The work is now presented in a balanced way and some previously hard-to-understand points have been clarified. The English of the new text sections is not quite on the same level as the English in the original manuscript. As this does not lead to ambiguity, technical editors of Nat. Comm. will probably fix it, but you should carefully check your proofs. Nowadays, you do not need a native English speaker for writing reasonable and correct English. Just translate your English text to Chinese by DeepL (or similar software), check that the Chinese version is fine, and translate it back to English.

Reply:

Thanks for the suggestion. We have carefully polished the language.